# A Putative Prophylactic Solution for COVID-19: Development of Novel Multiepitope Vaccine Candidate against SARS-COV-2 by Comprehensive Immunoinformatic and Molecular Modelling Approach

**DOI:** 10.3390/biology9090296

**Published:** 2020-09-18

**Authors:** Hafiz Muzzammel Rehman, Muhammad Usman Mirza, Mian Azhar Ahmad, Mahjabeen Saleem, Matheus Froeyen, Sarfraz Ahmad, Roquyya Gul, Huda Ahmed Alghamdi, Muhammad Shahbaz Aslam, Muhammad Sajjad, Munir Ahmad Bhinder

**Affiliations:** 1Institute of Biochemistry and Biotechnology, University of the Punjab, Lahore 54590, Punjab, Pakistan; muzzammel.phd.ibb@pu.edu.pk (H.M.R.); shahbaz.ibb@pu.edu.pk (M.S.A.); 2Department of Human Genetics and Molecular Biology, University of Health Sciences, Lahore 54590, Punjab, Pakistan; drmianazharahmad@gmail.com (M.A.A.); munirbhinder@uhs.edu.pk (M.A.B.); 3Department of Pharmaceutical and Pharmacological Sciences, Rega Institute for Medical Research, Medicinal Chemistry, University of Leuven, B-3000 Leuven, Belgium; muhammadusman.mirza@kuleuven.be (M.U.M.); mathy.froeyen@kuleuven.be (M.F.); 4Department of Health, Government of the Punjab, Lahore 54590, Punjab, Pakistan; 5Drug Design and Development Research Group (DDDRG), Department of Chemistry, Faculty of Science, University of Malaya, Kuala Lumpur 50603, Malaysia; sarfraz.ahmad@um.edu.my; 6Faculty of Life Sciences, Gulab Devi Educational Complex, Lahore 54590, Punjab, Pakistan; roquyya.gul@gdec.edu.pk; 7Department of Biology, College of Sciences, King Khalid University, Abha 61413, Saudi Arabia; hudaghamdi@kku.edu.sa; 8School of Biological Sciences, University of the Punjab, Quaid e Azam Campus, Lahore 54590, Punjab, Pakistan; sajjad.sbs@pu.edu.pk

**Keywords:** COVID-19, SARS-CoV-2, spike protein, multiepitope vaccine, molecular modeling

## Abstract

**Simple Summary:**

COVID-19, caused by a novel coronavirus, SARS-CoV-2, first emerged in China in December 2019, and then spread around the globe with more than 29 million confirmed infections. Immunoinformatics and molecular modelling techniques are time-efficient methods that are used to accelerate the discovery and design of the candidate peptides for vaccine development against SARS-COV-2. Recently, the use of multiepitope vaccines has proved to be a promising immunization strategy against different viruses and other pathogens. In the current study a comprehensive in silico strategy was used to design stable multiepitope vaccine construct (MVC) from B-cell and T-cell epitopes of essential SARS-CoV-2 proteins which include, spike, main protease, non-structural protein 12 (polymerase), and Nsp13 (helicase) with the help of adjuvants and linkers. Molecular dynamics studies revealed that the MVC displayed favourable molecular interactions with human Toll-like receptors (TLRs), which are known in triggering an innate and adaptive immune response. Furthermore, the MVC was checked for its recombinant production in Escherichia coli using a well-known expression system. The MVC showed a stable three-dimensional structure and could serve as a potential candidate for vaccine production, which warrant further experimental research for validation.

**Abstract:**

The outbreak of 2019-novel coronavirus (SARS-CoV-2) that causes severe respiratory infection (COVID-19) has spread in China, and the World Health Organization has declared it a pandemic. However, no approved drug or vaccines are available, and treatment is mainly supportive and through a few repurposed drugs. The urgency of the situation requires the development of SARS-CoV-2-based vaccines. Immunoinformatic and molecular modelling are time-efficient methods that are generally used to accelerate the discovery and design of the candidate peptides for vaccine development. In recent years, the use of multiepitope vaccines has proved to be a promising immunization strategy against viruses and pathogens, thus inducing more comprehensive protective immunity. The current study demonstrated a comprehensive in silico strategy to design stable multiepitope vaccine construct (MVC) from B-cell and T-cell epitopes of essential SARS-CoV-2 proteins with the help of adjuvants and linkers. The integrated molecular dynamics simulations analysis revealed the stability of MVC and its interaction with human Toll-like receptors (TLRs), which trigger an innate and adaptive immune response. Later, the in silico cloning in a known pET28a vector system also estimated the possibility of MVC expression in *Escherichia coli*. Despite that this study lacks validation of this vaccine construct in terms of its efficacy, the current integrated strategy encompasses the initial multiple epitope vaccine design concepts. After validation, this MVC can be present as a better prophylactic solution against COVID-19.

## 1. Introduction

The severe acute respiratory syndrome coronavirus 2 (SARS-CoV-2), an enveloped, non-segmented positive-sense RNA virus, causes severe respiratory infection [1]. The ongoing 2019–2020 outbreak of coronavirus disease 2019 (COVID-19) [2,3,4] has led to 727,435 deaths with 19,687,156 confirmed cases globally as per 10 August 2020, and the World Health Organization (WHO) has declared COVID-19 a global health emergency [5]. Coronaviruses are highly pathogenic viruses and are known to be contagious, which was revealed by the SARS and MERS (Middle East respiratory syndrome) outbreak in 2002 and 2012 [6,7]. The recent SARS-CoV-2 is considered as the seventh known human coronavirus (HCoV) from the same family after 229E, NL63, OC43, HKU1, MERS-CoV, and SARS-CoV [8].

Like other coronaviruses, SARS-CoV-2 is spherical, having a diameter of about 125 nm, and its genome (~30 kb) contains at least six open reading frames, which encode 16 non-structural proteins and 4 major structural proteins, namely, a spike protein (S), a form of glycoprotein; a membrane protein (M), which consists of the membrane; an envelope protein (E); and a nucleocapsid (N) protein, encoded by the ORFs near the 3′end of the genome. Among these structural proteins, the spike (S) glycoprotein binds to the cellular receptor angiotensin-converting enzyme 2 (ACE2), and is responsible for causing the viral infection [9]. The S precursor protein of SARS-CoV-2 can be proteolytically cleaved into S1 (685 amino acids) and S2 (588 amino acids) subunits [10]. Owing to the integral role of S protein between viral and host cell membrane interactions, it could be a potential target for developing new SARS-CoV-2 vaccines. Previous studies related to the development of anti-SARS-CoV vaccines and therapeutics that target S protein have already been reported [11,12,13,14]. Most of the non-structural proteins play an essential role in viral replication, mainly SARS-CoV-2 main protease (Mpro), also known as chymotrypsin-like protease (3CLpro) [15,16], Nsp13 helicase [17], and the Nsp12 RNA-dependent RNA polymerase [18]. These proteins are also highly conserved among coronaviruses [19].

Owing to the high mortality rate of patients, there is an urgent need to develop vaccines and anti-viral drugs to combat the COVID-19 outbreak. Although phenomenal efforts are in progress in developing vaccines and through repurposing studies [20,21,22], with the advancement in computational biology, it is now possible to accelerate the drug discovery pipeline and vaccine development [23,24,25], and these methods have surpassed the conventional methods [26,27]. Although Felipe and coworkers also reported a live attenuated vaccine using yellow fever 17D as a vector, that can induce SARS-CoV-2 neutralizing antibodies [28]. Numerous studies have been published related to the B- and T-cell epitope-based vaccine development using in silico immunoinformatics methods [29,30,31,32,33,34,35,36].

Keeping in view the urgency situation of the COVID-19 outbreak, developing an effective vaccine is, therefore, a prime research priority. There are some studies on the development of vaccines against SARS-CoV-2 and some vaccines reached have trial phase as well. It will still require 12 to 18 months to develop an effective vaccine [37]. The current study deals with the modelling of novel multiepitope vaccine for SARS-CoV-2 using a cost-effective integrated immunoinformatics approach. This approach has been proved to be promising against viral diseases caused by viruses like yellow fever [38], SARS-CoV [39], influenza [40], Zika [41], Congo Virus [42], and pathogens including *L. donovani* [43] and *S. pneumoniae* [44]. Few multiple epitope vaccine strategies proved to be effective against *H. pylori* infection in the BALB/c mice model [45,46], chronic hepatitis B virus infection [47], and foot-and-mouth disease virus (serotype A) in pigs [48].

In the present study, we effectively designed the multiepitope subunit vaccine construct (MVC) by considering the potential B- and T-cell epitopes of SARS-CoV-2 Spike, Mpro, Nsp-12 polymerase, and Nsp13 helicase proteins. The antigenicity, allergenicity, and physiochemical properties of B- and T-cell epitopes were also measured. Later, the structural analysis of MVC interaction with Toll-like receptors (TLRs) was analyzed through molecular dynamics (MD) simulations and binding free energies were estimated. TLRs establish an important link between innate and adaptive immunity. Engagement of TLR signaling pathways is a promising mechanism for accelerating vaccine responses and is involved in therapeutic immunization against infectious diseases [49]. Thus, the interaction of a multiepitope vaccine construct designed through an integrated modelling approach may trigger innate and specific adaptive immunity by activating TLR signaling pathways and may produce a promising immune response against SARS-CoV-2.

## 2. Materials and Methods

### 2.1. Coronavirus Protein Sequences and Structural Information

The primary amino acid sequences of SARS-CoV-2 main protease (Mpro) (306 amino acids), Nsp12 RNA dependent RNA polymerase (932 amino acids), spike (1237 amino acids), and Nsp13 helicase (601 amino acids) proteins were retrieved from GenBank ID: AHZ13508.1. For structural studies, the crystal structures of recently deposited SARS-CoV-2 Mpro (PDB ID: 6LU7) and spike (PDB ID: 6VYB) protein were obtained from PDB, while the homology models of SARS-CoV-2 Nsp12 RNA polymerase and Nsp13 helicase were obtained from our recent study [50]. These homology models were generated from templates that showed 99.83% and 96.08% identities with SARS-CoV Nsp12 (PDB ID: 6NUR) [18], Nsp13 (PDB ID: 6JYT) [17]. These models showed strikingly similar domain architecture with SARS-CoV and were found to be reliable enough to use in epitope identification studies.

### 2.2. Prediction of Linear and Conformational B-Cell Epitopes

The interaction between the antigenic B-cell epitope and B-lymphocyte causes the B-lymphocytes to differentiate into memory cells and antibody-secreting plasma [51]. B-cell epitope has two significant features, including accessibility to the flexible region and the hydrophilic nature of an immunogen [52]. As per the prediction of Parker hydrophilicity, for surface accessibility, Emini prediction [53], antigenicity scale for Kolaskar and Tongaonkar [54], and flexibility prediction for Karplus and Schulz [55], the analysis was employed arithmetically at IEDB (http://www.iedb.org/). Discontinuous (conformational) epitopes prediction for B-cell was performed using Ellipro from IEDB (http://tools.immuneepitope.org/toolsElliPro/) [56], which used three diverse algorithms such as residues’ protrusion index (PI) [57], adjoining clustering residues liable upon PI, and approximation of protein shape [58].

### 2.3. Prediction of Potential Cytotoxic T-Lymphocyte (CTL) Epitopes

NetCTL.1.2 server (http://www.cbs.dtu.dk/services/Net CTL) was used to predict the CTL epitopes [59], The eliciting of CTLs happened on the surface of antigen-presenting MHC (major histocompatibility complex) molecules. To assimilate the MHC class I binding, efficiency of TAP transport, and the cleavage of proteasomal C-terminal, NetCTL 1.2 server was employed. HLA (human leukocyte antigen) alleles and peptide lengths both were selected and submitted for prediction of T-cell epitopes as an output. For predicting the TAP transport efficiency, the weight matrix was utilized, while for cleavage of proteasomal C-terminal and MHC class I binding, the ANN (artificial neural network) was employed.

### 2.4. Epitope Prediction of Helper T-Cell

For the prediction of the epitope of helper T-cell, NetMHCII 2.2 Server was used, which gives a 15-mer epitope for human alleles. NetMHCII 2.2 Server uses an artificial neuron network for the prediction of a peptide with human alleles, that is, HLA-DP, HLA-DR, and HLA-DQ [60]. On the basis of receptor interaction, MHC II epitopes were predicted and deduced from IC50 values, as well as the assigned percentile ranks. The peptides that show a strong interaction have an IC50 value of <50 nM, while those having intermediate and low affinity have IC50 values of <500 and <5000, respectively. Therefore, the percentile rank has a direct relation with IC50 and inverts to the affinity for epitope.

### 2.5. Multiepitope Vaccine Designing

The MVC was designed by connecting the peptide sequences in a successive manner with the help of suitable linkers. The occurrence of overlapping residues amid the B-cell (BCL), HTL, and CTL epitopes was unwavering and epitopes with overlapping regions were used for multiepitope vaccine design. It has been established that human β-defensins have an important role in presenting the microbial peptides to antigen presenting cells and the inflammatory response, thus enhancing the immunogenicity of the bound antigen; therefore, β-defensins can be used as adjuvants [61,62,63]. Recently, mammals’ β-defensin was documented to have a possible role to confer HIV (human immunodeficiency virus) infection as a mucosal adjuvant; consequently, owing to its adjuvant characteristics against viral infection [64], it was chosen and added to the N- and C-terminal sequences of the vaccine construct. Adjuvants were joined with epitopes at the N- and C-terminal using the EAAAK linker, whereas intra-CTL epitopes were joined using the AAY linker. After the CTL epitope, HTL epitopes were added next to the CTL epitope using the GPGPG linkers, as used in a previous study [64].

### 2.6. Antigenicity and Allergenicity Estimation of the MVC

To be an effective and safe vaccine candidate, the vaccine candidate should be nonallergic with minimum off-targets effects. The nonallergenic and allergenic behaviors of the MVC were assessed by three servers, AllerTOP V2.0 (http://www.ddg-pharmfac.net/AllerTOP/), AlgPred (http://www.imtech.res.in/raghava/algpred/) and AllergenFP 1.0 [65]. Out of these, the latter categorizes the protein sequence (input) by a k-nearest neighbor algorithm (kNN; k = 3) on the basis of the training set comprising 2210 already known allergens from diverse species and nonallergens (*n* = 2210) from the similar species. The former assimilates the SVM module for the prediction of the allergenic nature of protein with high accuracy. The MAST/MEME allergen motif was examined with the help of MAST (Motif Allignment and Search Tool), and the allergenic nature was allocated if an identical motif was determined.

Evaluation of antigenicity of the MVC was done using two freely available servers, VaxiJen v2.0 (http://www.ddg-pharmfac.net/vaxijen/VaxiJen/VaxiJen.html) [66] and ANTIGENpro (http://scratch.proteomics.ics.uci.edu/), where the latter categorizes the antigen based only on the physio-chemical characteristics of the input protein sequence instead of the sequence placement algorithm. The correctness of the server is relatively high and differs between 70% and 89% based on the target organism. While the former envisages the entire protein antigenicity based on the results obtained by the protein microarray data analysis, it predicts the antigenicity as being independent of the pathogen, but this approach is sequence-based.

### 2.7. Physiochemical Parameters Evaluation

The vaccine construct sequence was used in ProtParam (http://web.expasy.org/protparam/) [67] to examine its physiochemical properties. The criteria on which the sequences of multiepitope vaccine were examined are theoretical PpI, half-life, instability index, aliphatic index, grand average, and stability profiling of hydropathy.

### 2.8. Tertiary Structure Prediction and Refinement of MVC

The final vaccine construct was submitted to structure prediction server known as RaptorX (http://raptorx.uchicago.edu/StructurePrediction/predict/) [68] and I-TASSER [69]. It is an exceptional server for protein 3D structures’ predictions on the ab initio method and is able to generate from the template that lies in the twilight zone (<30%). It utilizes an exclusive nonlinear context-specific alignment and prospective consistency algorithm. The generated models were evaluated through MolProbity for all-atom contacts and geometry [70]. The model was selected for further refinement through molecular dynamics simulations that showed reliable Ramachandran evaluations.

### 2.9. Stability Enhancement of MVC by Disulfide Engineering

Before moving towards the docking protocol, it is essential to enhance the stability of the model through disulfide engineering. It is a novel concept to introduce disulfide bonds to the modelled protein structure. Consequently, the multiepitope model was employed to the Disulfide by Design 2.0 server [71] to achieve disulfide engineering. The protein model was uploaded to identify the residue pairs, which can be utilized for disulfide engineering. To create the disulfide bonds, four residues were selected to mutate them with cysteine residue by the Disulfide by Design 2.0 server.

### 2.10. Molecular Docking of Vaccine Constructs with TLR4

In order to analyze the binary interaction of MVC with TLRs, protein–protein docking was performed using Cluspro [72,73].The PDB structures of TLR-4 (PDB: 4G8A) and TLR-3 (PDB: 1ZIW) were retrieved from PDB. The multiepitope vaccine model was used as a ligand. Both proteins were prepared accordingly, by removing heteroatoms, and the addition of hydrogens and charges. ClusPro is an automated protein–protein docking server, which generates root mean square deviation (RMSD)-based clustering of 1000 docked conformations. Each representative model is chosen from a cluster of docked models based on the scoring function it uses. The most representative docked conformation from the largest cluster was used for further structural analyses.

### 2.11. Molecular Dynamics Simulation for TLRs/MVC Complex

Molecular dynamics simulations were performed in two steps: (1) a 50 ns MD simulation was performed to optimize and refine 3D model of multiepitope vaccine construct before docking, and (2) a second 50 ns MD simulations to examine the backbone stability of TLRs/MVC complexes. After MD, MD clustering was performed, which typically takes the representative conformation from the largest cluster (within 1 Angstrom deviation). These clusters are generated based on the deviation over the course of total snapshots (in our case, after every 2 ps, which generated a total of 25,000 snapshots). This analysis considers all meaningful conformations over the time of simulation. All simulations were performed by AMBER simulation package 18 using the same protocol as described in previous immunoinformatics studies [30,32]. Briefly, with stepwise minimization and an equilibration procedure, the solvated system in explicit water molecules (TIP3P) was submitted to a production run at standard temperature (300 K) and pressure (1 bar). The trajectories were collected after every 2 ps for a complete production run, and the CPPTRAJ module was utilized to analyze trajectories. The MD simulation complexes were analyzed using Chimera 1.14.

### 2.12. Codon Adaptation and In Silico Cloning

The sequence of multiepitope vaccine construct was employed to the online server JCat for reverse translation, and cDNA was obtained, which was submitted for codon optimization [74]. The cDNA was evaluated by codon optimization according to the codon adaptation index (CAI) and GC content of the sequence. The acceptable range of the GC content is 30–70% and the value of CAI varies from 0 to 1. The higher value of CAI indicates a higher level of gene expression [75]. The maximum value of CAI is 1 and is considered ideal, whereas a value of more than 0.8 is also acceptable. After this step, the adapted and optimized sequence of the nucleotides consistent to the design of multiepitope vaccine construct was cloned using the restriction cloning module of SnapGene toll in the vector pET28a (+) of *E. coli*.

### 2.13. In Silico Immune Simulation

To check the immunogenic potential of the vaccine construct, an in silico immune simulation approach was employed using the C-immsim server [76]. The position specific scoring matrix approach was used by the server for the analysis. The server used three compartments of mammals for immune stimulation, that is, lymph node, thymus, and bone marrow [77]. The defaults constraints for simulation were employed, which are as follows: simulation volume (10), simulation steps (100), random seed (12345), host HLA selection (MHC Class I A0101 allele, B MHC class I B0702, DR MHC class II DRB1_0101 allele), and the time for the injection was set as 1.

## 3. Results

In the present research, plausible T-cell and B-cell epitopes (discontinuous and continuous) from SARS-CoV-2 Spike, Mpro, Nsp12 RNA polymerase (RdRp), and Nsp13 helicase proteins were recognized to design peptide vaccines to counter SARS-CoV-2 infection. Most potential epitopes were selected and joined together with appropriate linkers and adjuvant. The 3D model was generated using various online servers, and a reliable model was used for docking and MD simulation studies. Docking and immuno-informatics method are helpful for the prediction of the binding interaction between TLRs and ligand (multiepitope vaccine) complexes, and analysis was done as these are proven useful tools in identifying novel multiepitope vaccines [23,32].

### 3.1. Antigenic B-Cell Epitope Prediction

Depending on the physicochemical properties of amino acids, which have already been observed in practically determined antigen-based epitopes, Kolaskar and Tongaonkar’s approach was used for predicting antigenic epitopes of provided sequences. Seventy-five percent experimental precision has been reported for this approach [54]. Using this method, 11 antigenic peptides with 9–14 amino acid length were observed, including two heptapeptides from SARS-CoV-2 Mpro (Table 1). Likewise, out of 932 amino acids, 37 antigenic peptides were predicted in SARS-CoV-2 Nsp12 polymerase. For RdRp, the length of the antigenic peptides was 6–29 amino acid along, with ten heptapeptides and nine octapeptides (Table 2). For Nsp13 helicase, 18 antigenic peptides were predicted, and the length of the antigenic peptide was 6–38, with 7 hexapeptides (Table 3). For the spike protein, 46 antigenic peptides were predicted from 1273 amino acids (Table 4).

Moreover, Kolaskar and Tongaonkar’s approach also projected the highest residual score of each amino acid in all investigated proteins. In SARS-CoV-2 Mpro, 211 out of 306 amino acids have greater than 1.000 residual scores. From position 85 to 91, the antigenic peptide (CVLKLKV) having lysine at the 88th position was identified with a maximal residual score of 1.22. The Nsp12-RdRp has 686 residues out of 932 with a residual score above 1.000, and valine at position 473, in the peptide (LFVVEVV) from 470 to 476, has a maximum residual score of 1.246. Likewise, for Nsp13 helicase protein, 479 out of 601 amino acids were predicted with a residual score greater than 1.000, and lysine present at the 28th position in an antigenic peptide from 25 to 31 (LCCKCCY) has a maximum residual score of 1.284. For SARS-CoV-2 S protein, 958 out of 1273 amino acids were predicted to have a residual score higher than 1.000, and leucine at position 8 of the antigenic peptide from position 5 to 11 (LVLLPLV) showed a maximal residual score of 1.261.

A graphical depiction of peptides predicted for B cell from investigated SARS-CoV-2 proteins based on sequence position along the x-axis and antigenic propensity (AP) as the y-axis is shown in Appendix A. Divergence in AP is related to the length of the sequence. The minimum AP score for Mpro was 0.844 and the maximum AP score was 1.22 (A), while the maximum and minimum AP scores of Nsp12, Nsp13, and S protein were 1.246, 1.284, and 1.261, respectively, and 0.858, 0.893, and 0.866, respectively.

### 3.2. Prediction of Cytotoxic T-Lymphocyte (CTL) Epitopes

An infected cell having antigen-presentation triggers the T-cell to turn out as an effector cell and kill the infected cells. Cell death or self-destruction is detected after the attack of CTLs on effected cells. The pathogen’s peptide fragment and molecule of MHC interact and are exposed on the cell surface of infected cells. CTLs identify the complex of peptide–protein; moreover, as a consequence, infected cells are killed. The processing of fragment of the peptide (antigen), along with its appearance to the T-cell, is achieved through different steps. Peptides are treated in the cytoplasm through proteasome and transferred to the endoplasmic reticulum (ER) later, where MHC is produced. Here, the peptide is transported to the MHC I molecule by the transporter associated with antigen processing (TAP). Afterwards, a complex of peptide–MHC-I is transferred to the surface of the cell. A varied array of peptides is attached to each allelic type of MHC-I protein. The molecule of MHC can interact with peptides strongly as the pathogens attempt to mutate the MHC molecule’s epitope. Therefore, the MHC molecule displays strong binding with a diversity of peptides [78].

Prediction of CTL epitope is a significant in silico tool in designing the vaccine as it decreases the time and necessity for in vitro trials. NetCTL 1.2 server [59] was used for the prediction of CTL epitope. For all investigated SARS-CoV-2 proteins, the peptide sequences were predicted as CTL epitopes based on three main factors, which include their MHC binding capacity, proteasomal cleavage of the peptide from C-terminal, and affinity for the TAP transporter with the default threshold prediction score being >0.75000. Among all the peptides, 11 peptide sequences from S protein, 4 peptides from Mpro, 19 from Nsp12, and 10 from Nsp13 were selected as CTL epitopes. These CTL epitopes were also predicted as an antigenic site. Hence, these peptides can be considered as potential vaccine candidates (Table 5, Table 6, Table 7 and Table 8). A complete list of peptides for these four proteins is also given in Appendix A.

### 3.3. Structure-Based Epitope Prediction

ElliPro was used to predict the epitopes from the 3D structure of proteins [56]. This advanced program is web-based and used to study the correlation among antigenicity, flexibility, and solvent accessibility of proteins’ structure. Furthermore, differentiation of predicted epitopes based on interactions of protein–antibody is an essential property of this program. ElliPro measures the PI score (protrusion index), which shows the percent of atoms of proteins that spread beyond the molecular mass/bulk as well as those responsible for binding antibodies. On the basis of the PI score (>0.7), five, three, and two discontinuous epitopes were selected for SARS-CoV-2 S, Nsp13 helicase, and Nsp12 polymerase, respectively, while only one epitope was identified for Mpro that showed PI > 0.7. The graphical illustration of discontinuous epitopes is shown in Figure 1, while number of residues and epitope scores are tabulated in Table 9, Table 10, Table 11 and Table 12.

### 3.4. Epitope Prediction for (HTL) Helper T Lymphocytes

MHC class II epitope, which shows high binding affinity, was predicted for human alleles HLA-DP, HLA-DQ, and HLA-DR based on their IC50 values from Net MHC II 2.2 server. These epitopes were described as HTL epitopes. The epitopes with similar sequences were overlapped to get a single epitope. A total of 21 high binding HTL epitopes were selected for the spike (S), main protease, RdRp, and helicase for the novel multiepitope vaccine (Appendix A).

### 3.5. Design and Construction of Final Multiepitope Vaccine

The overlapped and high scoring CTL and HTL epitopes found from SARS-CoV-2 S, Mpro, Nsp12 polymerase, and Nsp13 helicase were combined to form the multiepitope vaccine construct (MVC). To increase the immune response, human β-defensin 2 (hβD-2) (PDB ID: 1FD3), the sequence of GIGDPVTCLKSGAICHPVFCPRRYKQIGTCGLPGTKCCKKP and hBD-3 (PDB ID:1KJ6), the sequence of GIINTLQKYYCRVRGGRCAVLSCLPKEEQIGKCSTRGRKCCRRKK were selected as adjuvants at the N- and C-terminals sequence of the vaccine construct, respectively, with linker EAAK [79,80]. After the adjuvant CTL epitopes were combined using appropriate AAY linkers, HTL epitopes were joined together with GPGPG linkers [81] as displayed in Appendix A. By combining potential CTLs, HTLs epitopes, and adjuvants, a multiepitope vaccine construct of 1057 amino acids was constructed.

### 3.6. Parametric Evaluation of Physiochemical Properties

By estimating the multiepitope vaccine construct (MVC) using ProtParam server to estimate physicochemical properties [67], it was found that MVC weighed 114.6 kDa. The hypothetical isoelectric point (pI) was 8.15, displaying the basic nature of the MVC, and the assessed in vitro half-life was 30 h in mammals’ reticulocytes [82]. The assessed half-life indicates the time acquired by the protein to remain half of the quantity as originally produced in the cell. The instability index was also predicted to be 35.56 and classified the MVC as stable in nature. The aliphatic index [83] was also examined, which displays the relative volume retained by the aliphatic side chain. It might be reflected as a positive variable for the extension of the thermostability of globular proteins. The attained values of the aliphatic index were found to be 80.93, indicating that, at varied temperatures, the protein is thermostable. The grand average value of Hydropathy [84] signifies the summation of the hydropathy rate and, along with sequence of amino acid, indicates the hydrophilic and hydrophobic nature of the protein. The observed grand average value of Hydropathy for the vaccine protein was found to be 0.158.

### 3.7. Assessment of Allergenicity and Immunogenicity

The designed subunit of the vaccine was assessed on the allergenic parameter through AllergenFP 1.0 and AlgPredand AllerTOP 2.0 servers. All these servers predicted the non-allergenic nature of MVC. The antigenicity connected to the vaccine subunit was projected through VaxiJen v2.0 servers. According to the outcome of VaxiJen, the antigenicity of the vaccine was 0.4259, displaying it as a plausible antigen. Thus, the attained outcome from servers exhibited a high possibility of the subunit vaccine’s antigenic and non-allergenic nature.

### 3.8. Structure Prediction and Validation of MVC

In order to analyze the 3D confirmation of MVC, the 1073 amino acid peptide sequence was utilized for the prediction of the 3D model. Multiple softwares were used for modeling, including RaptorX [85] and I-TASSER [86,87], in order to avoid biases. The information of the secondary structure result showed 34% helical, 19% E, and 45% coiled assembly. For homology modeling, the *p*-value is a good parameter to describe the relative quality of the model and a lower *p*-value indicates that the quality is good for the modeled structure. The *p*-value obtained for the MVC structure was 1.29 × 10^-6^, which is lower and significant. For I-TASSER modeling, the model with the highest C-score was selected. The models generated from both servers were compared and validated through the MolProbity server [70]. The structure with a better M-score (which combines the clash score, rotamer, and Ramachandran estimations into a single score) was utilized for extensive refinement through MD simulations. The refinement included optimizing bond lengths and angles and removing clashes in geometry [88,89,90]. The root mean square deviation (RMSD) was calculated for 50 ns. Figure 2A displays the MD refined 3D-model of MVC and the all-backbone RMSD trajectory is shown in Figure 2B. Initially, the RMSD trajectory of MVC model gradually expanded until 30 ns and reached a value of ~9.5 Å. Later, the RMSD value continued to converge until 50 ns, with a deviation <1 Å. This higher RMSD of the simulated model indicated protein expansion during simulation to attain a more stable conformation. The averaged conformation of the MVC model was extracted from the trajectory and compared with the initial model through Ramachandran evaluations. The MD optimized MVC model showed that 86.2% (924/1073) of all residues were in Ramachandran favoured (>98%) regions, while the initial homology model showed only 72.49% (778/1073 residues) in Ramachandran favoured regions. Moreover, residues placed in Ramachandran favoured regions (>99.8%) increased from 83.8 (901/1073 residues) with 172 (16.02%) outliers to 95.5% (1024/1073) with 49 outliers (4.1%) (Figure 2C).

### 3.9. Disulfide Engineering for Vaccine Stability

Disulfide engineering was done to stabilize the modelled structure of MVC, by Disulfide by Design v2.0 server [71]. In the evaluation based on other parameters like Chi3 and energy value, only 07 residues pairs were selected as their value came under the permissible range, that is, energy value must be smaller than 2.2 and Chi3 must be between −87 and +97 degrees. Hence, a total of eight mutations were formed at the pairs of residues, named VAL6-ALA157, TYR138-ALA163, VAL360-GLY730, LEU462-TYR474, ALA499-ARG519, SER814-GLY923, GLY816-SER927, and THR934-GLY946.

### 3.10. Molecular Docking of Vaccine Constructs with TLR3 and TLR4

Molecular docking is the best in silico approach to finding out interactions between protein–protein and protein–ligand complexes [91,92,93,94]. Molecular docking of MVC with TLR4 and TLR3 receptors was performed using ClusPro 2.0, and 30 models were produced [95]. Among these, the complex with the lowest energy was selected. The energy scores attained for TLR3 and TLR4 were –1327.2 and –1270.2, respectively, and subjected to MD simulations to analyze the complex stability. The interaction profile of TLR3 and TLR4 with the MVC showed significant interactions, including H-bonds, salt bridges, and disulfide contacts (Appendix A). Both hydrogen bonds and salt bridges are particularly important in determining binding specificity [96]. It was observed that MVC established 16 H-bonds with TLR3 and 12 H-bonds with TLR4 within the range of 3.00 Å (Appendix A).

### 3.11. Molecular Dynamics Simulation for TLRs/MVC Complex

The stability of the TLRs/MVC docked complexes was further investigated by performing MD simulation for a period of 50 ns in an explicit solvent environment at 300 K. The potential energy of the simulation system was also found to be stable throughout the simulation period (data not provided). The MD refined MVC was utilized for docking, and both complexes showed relatively stable RMSD as compared with MVC alone (Figure 2). In the beginning, the MVC experienced small fluctuations, but remained interacted with the hydrophobic groove of TLR4 (Figure 3A) and TLR3 (Figure 3B), and showed consistent stability in the last ~25 ns. The radius of gyration (RoG) and solvent accessible surface area (SASA) analyses were achieved to determine the compactness [97,98] and protein solvent accessible surface area [99] of TLR3 and TLR4 and designed MVC throughout the MD run (Figure 3C–F). The results suggested similar trends in both complexes. The RoG plot (measured in nm) showed no conformational shift, except for small deviations that were evident owing to the flexible linkers utilized, and the overall structure remained stable between 31.5 and 33 nm. The compactness of TLR3 and TLR4 complexed with MVC suggested a strong binding interaction with the designed MVC. A similar description was revealed through SASA analysis (measured in nm^2^), representing the solvent accessible protein surface and its placement through folding, creating the adjustments in the exposed and buried regions of the surface area of proteins. SASA trajectories in both systems also showed a similar trend throughout the simulation period. The presented analysis suggested a stable structure with a significantly strong binding interaction with the vaccine construct, hence providing insights into the biological system’s stability [97,98].

### 3.12. Codon Adaptation and In Silico Cloning of the MVC

The reverse translation and codon optimization were performed for the sequence of MVC by the online JCat server [74]. The GC content and codon adaptation index (CAI) were determined out as output from the server. The GC content obtained for MVC was 54.39%, which lies in the acceptable range, that is, from 30% to 70%. Meanwhile, CAI value of MVC was 1, which indicates a high level of expression in the K12 strain of *E. coli*. Later, the restrictions sites of *NdeI* and *XhoI* were added and the MVC sequence was cloned in the pET28a (+) vector (Figure 4). The MVC sequence is represented in yellow with the restriction sites. The sequence of multiepitope vaccine construct was cloned between the 6-histidine residue on both sites, which will help in the purification of MVC.

### 3.13. Immune Simulation by MVC

The immune simulation response of MVC was determined by C-Immsim server. The MVC generated strong primary responses. It has been shown that the titer scale of combined antibodies, IgM and IgG, is approximate to 10,000/mL, and for the antibody, IgM is close to 7000 titer per ml (Figure 5A). The high level of immunoglobulin accomplishments was distinct with associated antigen reduction in both secondary and tertian responses. The level of soluble cytokine, interferon-gamma (IFN-g) was retained, and it was more than 400,000 ng/mL against the antigen, as shown in Figure 5B.

## 4. Discussion

The announcement of emergency by the World Health Organization (WHO) on the COVID-19 outbreak urged researchers to develop therapeutics, mainly the identification of drug candidates or vaccines [20]. The use of cost-effective and less time-consuming methods, especially immunoinformatics approaches, haas already assisted the researchers to predict potential antigenic epitopes for the multiepitope-based vaccine [38,39,40,44,75,100,101,102]. The multiple epitope vaccine has a distinctive design concept compared with classical single-epitope based vaccines [101,103,104,105]. The concept behind the scanning of the viral genome to find immunogenic epitopes leads to an elicited immune response without any reversal of viral pathogenesis [106].

To design a multiepitope vaccine, the research focused on the identification of epitopes for potential B and T cells using the immune-informatics approach. An in silico method can be employed using patho-genomics analysis on the genome on a vast scale to identify new vaccines [106,107]. Various limitations are there in the context of appropriate candidate antigens, their immunodominant epitopes, and experimental methods, which include the development of an effective delivery system [108,109]. Investigation of the whole spectrum of probable antigens is achievable through immunoinformatics and with the aid of molecular modelling to analyze the potential binding with host proteins [30,32,38,41,109]. Besides, the difficulty of culturing the pathogens as well as in vitro antigen expression problems can be avoided [102,110]. Some multiple epitope vaccines showed in vivo efficacy with promising protective immunity [45,46,48,103], while some have entered into phase-I clinical trials, including H2NVAC in patients with HER2-expressing ductal carcinoma in situ (DCIS) (NCT03793829), E1602 for patients with metastatic melanoma [111], EMD640744 in patients with advanced solid tumors [104], and TAB9 in non-HIV-1 infected human volunteers [112]. However, designing an effective multi-epitope vaccine remains a great challenge. Hence, estimation of B cell and CTL cell epitopes by different immune-informatics methods is considered to be a vital tool for designing a multi-epitope construct.

In the present research, potential T-cells and B-cell epitopes (discontinuous and continuous) were recognized from SARS-CoV-2 main protease, Nsp12 RNA polymerase, spike, and Nsp13 helicase proteins to design multi-epitope construct (MVC) using adjuvants (hβ defensins) and appropriate linkers. The employed linkers (GPGPG and AAY) were carefully selected because their length, composition, and structure may affect the activity of the domains and overall characteristics of the molecule [113]. For example, as being somewhat basic antigenic domains (isoelectric point pI > 8), a linker that contains more basic amino acids may increase the pI, such as KK [114]. Therefore, basic linkers were avoided and a glycine-rich linker, that is, GPGPG, was chosen for joining potential epitopes that usually improve solubility and allow the adjoining domains to be accessible and act freely [115]. Following this, a reliable MVC model was generated through molecular modeling and optimized accordingly. All-atom backbone stability of MVC was analyzed through molecular dynamics simulation over a period of 50 ns, because the optimal structural stability of MVC is considered a prime aspect in its efficacy [116], and to the trigger immune response by interacting Toll-like receptors (TLRs) signaling, as successful immunization results are accomplished through stimulation of the TLRs [49]. The resulting model showed fewer outliers, while rotamers were adjusted during the simulation. Molecular docking with TLR3 and TLR4 followed by 50 ns MD simulation revealed stability in the overall complex in the last ~20 ns. The designed MVC interacted with TLR3 and TLR4 directly and their molecular interactions were strengthened during MD simulation, which led to reducing the backbone RMSD fluctuation in both TLR/MVC complexes (Figure 3). However, the epitopes were estimated as non-allergenic, showed antigenicity, and predicted cloning in vector pET28a (+) of E. coli, but given the limitation of in silico tools, the expression and efficacy of the designed multiple vaccine construct should be further proven through in vitro and in vivo experiments.

## 5. Conclusions

COVID-19, after its first emergence in December 2019, widely spread to around 105 countries and the World Health Organization declared it as pandemic. This state of emergency urged to look for effective vaccine candidates and antiviral drugs. The immunoinformatics approach is fast and cost-effective to design and validate the candidate vaccines against such pathogens. In this study, a multiepitope vaccine using spike, Mpro, Nsp-12 polymerase, and Nsp13 helicase proteins of SARS-COV-2 was designed. The epitopes that can induce B- and T-cell mediated immune response were used to build the 3D model of the multiepitope vaccine, which was further validated for its stability and allergenicity. Molecular docking followed by molecular dynamics simulations of MVC with TLR3 and TLR4 was performed, which showed stable interactions of the candidate vaccine with these receptors. Overall, the MVC showed an overall stable structure and could serve as a potential candidate for vaccine production. Although present research is based on an integrated computational approach, further experimental research will be required to validate the effectiveness of the designed vaccine construct.

## Figures and Tables

**Figure 1 biology-09-00296-f001:**
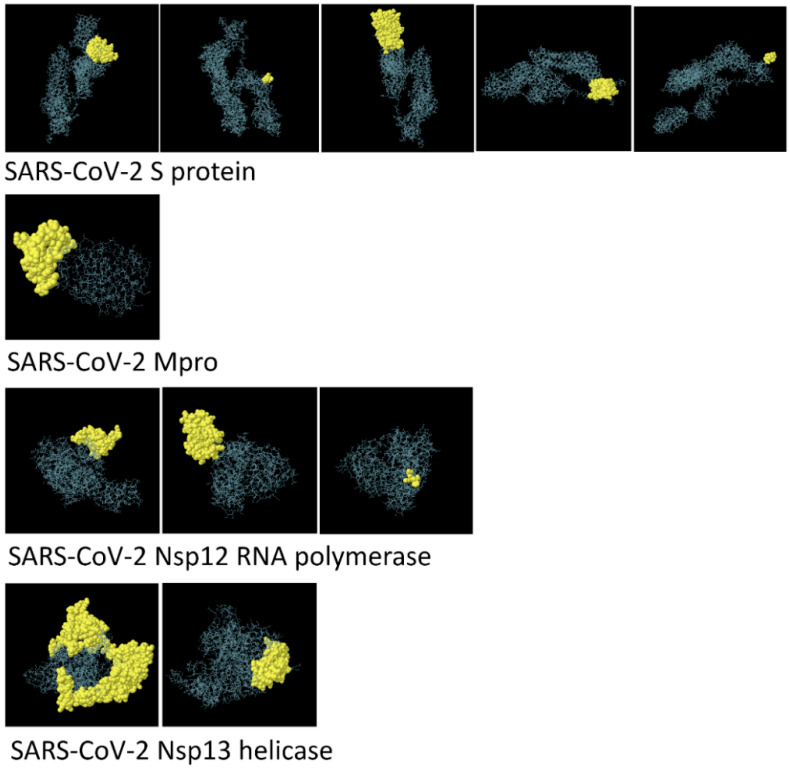
3D representation of discontinuous epitopes of severe acute respiratory syndrome coronavirus 2 (SARS-CoV-2) spike, Mpro, Nsp12 RNA polymerase, and Nsp13 helicase.

**Figure 2 biology-09-00296-f002:**
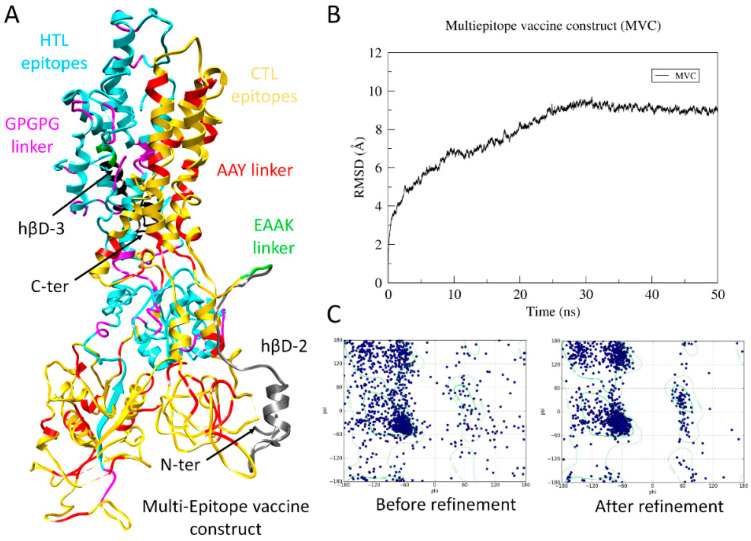
Molecular modeling of vaccine construct. (**A**) Structural representation of multiepitope vaccine construct (MVC) is displayed with regions (helper T lymphocytes (HTL), cytotoxic T-lymphocyte (CTL) epitopes, linkers, and adjuvants) highlighted accordingly. (**B**) Root mean square deviation trajectory (RMSD) of MVC analyzed over a period of 50 ns molecular dynamics (MD) simulations. (**C**) Ramachandhran evaluations of MVC before and after refinement through MD simulations.

**Figure 3 biology-09-00296-f003:**
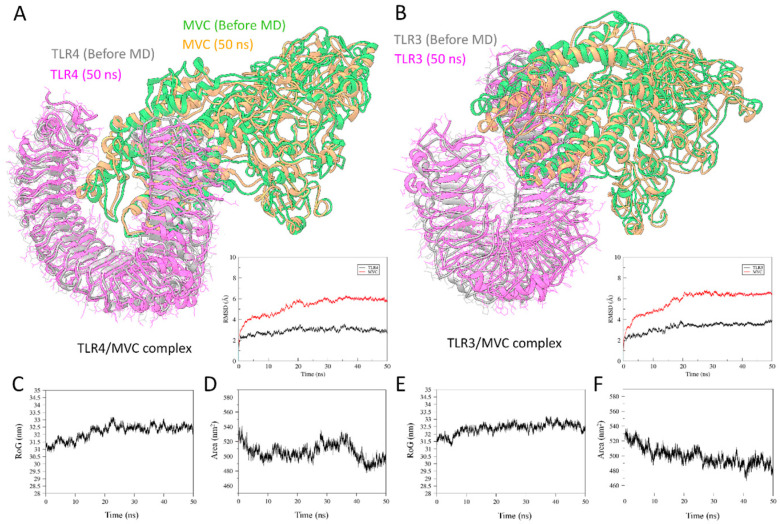
Toll-like receptor (TLR) complexed with a multiepitope vaccine construct (MVC). (**A**) Conformation of TLR4/MVC and (**B**) TLR3/MVC complex before and after 50 ns MD simulations, together with the RMSD plot at the bottom indicating the all-atom backbone deviation of TLR (in black) and MVC (in red). (**C**) Plot of radius of gyration (RoG) and (**D**) solvent-accessible surface area of TLR4/MVC complex throughout 50 ns MD simulation and TLR3/MVC (**E**,**F**).

**Figure 4 biology-09-00296-f004:**
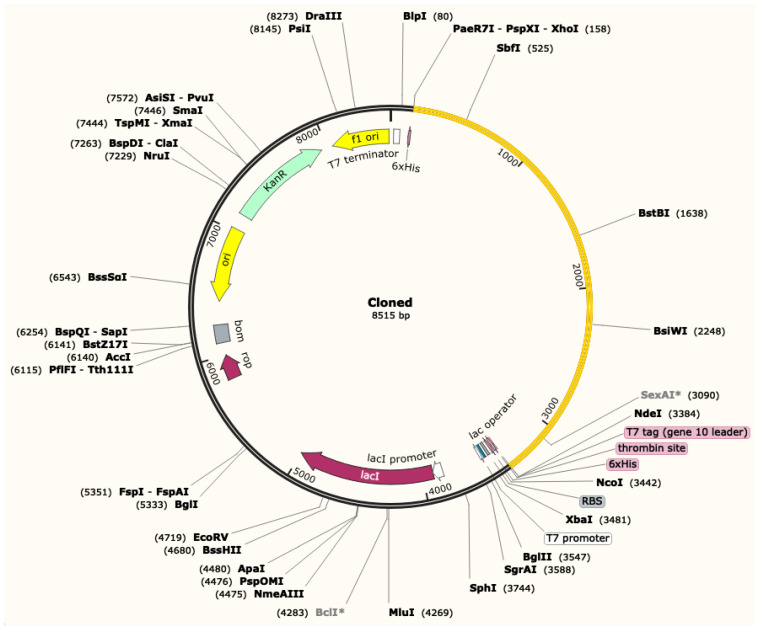
In silico cloning of the multiepitope vaccine construct (MVC). The cDNA of the MVC (yellow) was inserted at the upstream of the T7 promoter.

**Figure 5 biology-09-00296-f005:**
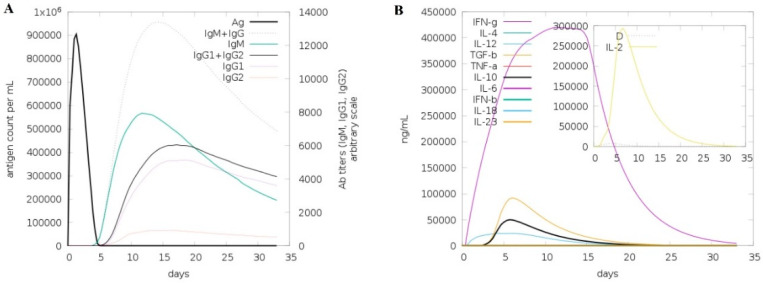
Computational immune simulation by C-Immsim using MVC as antigen. (**A**) Immunoglobulin/antibodies titer in response to antigen injection. (**B**) Production of interleukin (IL) and cytokines in response to antigen.

**Table 1 biology-09-00296-t001:** Predicted antigenic B-cell epitopes of severe acute respiratory syndrome coronavirus 2 (SARS-CoV-2) main protease (Mpro).

No.	Start	End	Peptide	Length
1	15	23	GCMVQVTCG	9
2	32	45	LDDVVYCPRHVICT	14
3	65	72	NFLVQAGN	8
4	83	91	QNCVLKLKV	9
5	101	107	YKFVRIQ	7
6	111	120	TFSVLACYNG	10
7	123	129	SGVYQCA	7
8	153	162	DYDCVSFCYM	10
9	201	212	TVNVLAWLYAAV	12
10	244	253	QDHVDILGPL	10
11 *	258	271	GIAVLDMCASLKEL	14

***** 11 antigenic sites were predicted from the main protease. The underlined residues were also predicted as cytotoxic T-lymphocyte (CTL) epitope.

**Table 2 biology-09-00296-t002:** Predicted antigenic B-cell epitopes of SARS-CoV-2 Nsp12 RNA polymerase.

No.	Start	End	Peptide	Length
17	395	400	CFSVAA	6
3	50	56	KTNCCRF	7
8	171	177	ILRVYAN	7
10	201	207	IVGVLTL	7
13	327	333	GPLVRKI	7
20	557	563	VAGVSIC	7
21	573	579	QKLLKSI	7
22	585	591	ATVVIGT	7
26	670	676	GGSLYVK	7
28	725	731	HRLYECL	7
31	773	779	QGLVASI	7
2	28	35	TDVVYRAF	8
6	125	132	ADLVYALR	8
15	350	357	ELGVVHNQ	8
16	369	376	KELLVYAA	8
18	435	442	VELKHFFF	8
23	633	640	MASLVLAR	8
29	744	751	EFYAYLRK	8
32	783	790	KSVLYYQN	8
34	825	832	DYVYLPYP	8
4	67	75	DSYFVVKRH	9
25	658	666	ECAQVLSEM	9
30	760	768	DDAVVCFNS	9
35	839	847	GAGCFVDDI	9
36	859	867	FVSLAIDAY	9
1	8	17	LNRVCGVSAA	10
27	694	703	FNICQAVTAN	10
33	810	819	HEFCSQHTML	10
7	144	154	EILVTYNCCDD	11
9	183	193	RQALLKTVQFC	11
14	335	345	VDGVPFVVSTG	11
24	643	653	TTCCSLSHRFY	11
5	87	99	YNLLKDCPAVAKH	13
37 *	878	890	ADVFHLYLQYIRK	13
19	466	482	IRQLLFVVEVVDKYFDC	17
11	230	248	GVPVVDSYYSLLMPILTLT	19
12	295	323	HPNCVNCLDDRCILHCANFNVLFSTVFPP	29

***** 37 antigenic sites were predicted. The underlined residues were also predicted as CTL epitope.

**Table 3 biology-09-00296-t003:** Predicted antigenic B-cell epitopes of SARS-CoV-2 Nsp13 helicase protein.

No.	Start	End	Peptide	Length
3	70	75	YYCKSH	6
5	207	212	DAVVYR	6
10	369	374	DIVVFD	6
11	384	389	LSVVNA	6
13	423	428	NSVCRL	6
15	493	498	IGVVRE	6
17	542	547	DYVIFT	6
12	394	400	KHYVYIG	7
16	522	528	ASKILGL	7
18 *	570	576	VGILCIM	7
1	4	11	ACVLCNSQ	8
6	222	230	GDYFVLTSH	9
9	353	361	EQYVFCTVN	9
4	78	87	PISFPLCANG	10
14	449	458	VDTVSALVYD	10
7	237	250	APTLVPQEHYVRIT	14
8	292	325	AIGLALYYPSARIVYTACSHAAVDALCEKALKYL	34
2	21	58	RRPFLCCKCCYDHVISTSHKLVLSVNPYVCNAPGCDVT	38

***** 18 antigenic sites were predicted. The underlined residues were also predicted as CTL epitope.

**Table 4 biology-09-00296-t004:** Predicted antigenic B-cell epitopes of SARS-CoV-2 spike protein.

No.	Start	End	Peptide	Length
1	4	18	FLVLLPLVSSQCVNL	15
2	34	41	RGVYYPDK	8
3	44	51	RSSVLHST	8
4	53	60	DLFLPFFS	8
5	65	70	FHAIHV	6
6	81	87	NPVLPFN	7
7	115	121	QSLLIVN	7
8	125	134	NVVIKVCEFQ	10
9	136	146	CNDPFLGVYYH	11
10	168	174	FEYVSQP	7
11	210	216	INLVRDL	7
12	223	230	LEPLVDLP	8
13	239	248	QTLLALHRSY	10
14	263	270	AAYYVGYL	8
15	272	278	PRTFLLK	7
16	288	295	AVDCALDP	8
17	333	339	TNLCPFG	7
18	359	371	SNCVADYSVLYNS	13
19	376	385	TFKCYGVSPT	10
20	430	435	TGCVIA	6
21	488	495	CYFPLQSY	8
22	505	527	YQPYRVVVLSFELLHAPATVCGP	23
23	592	599	FGGVSVIT	8
24	607	615	QVAVLYQDV	9
25	617	627	CTEVPVAIHAD	11
26	647	653	AGCLIGA	7
27	667	674	GAGICASY	8
28	687	693	VASQSII	7
29	723	730	TTEILPVS	8
30	735	741	SVDCTMY	7
31	750	763	SNLLLQYGSFCTQL	14
32	781	788	VFAQVKQI	8
33	803	808	SQILPD	6
34	837	843	YGDCLGD	7
35	847	853	RDLICAQ	7
36	858	864	LTVLPPL	7
37	873	880	YTSALLAG	8
38	959	966	LNTLVKQL	8
39	973	979	ISSVLND	7
40	1003	1011	SLQTYVTQQ	9
41	1030	1037	SECVLGQS	8
42	1057	1070	PHGVVFLHVTYVPA	14
43	1079	1085	PAICHDG	7
44	1123	1132	SGNCDVVIGI	10
45	1174	1179	ASVVNI	6
46 *	1221	1256	IAGLIAIVMVTIMLCCMTSCCSCLKGCCSCGSCCKF	36

***** 46 antigenic sites were predicted. The underlined residues were also predicted as CTL epitope.

**Table 5 biology-09-00296-t005:** Predicted CTL from SARS-CoV-2 S protein *. TAP, transporter associated with antigen processing.

Residue Number	Peptide Sequence	Predicted MHC Binding Affinity	Rescale Binding Affinity	C-Terminal Cleavage Affinity	TAP Transport	Prediction Score	MHC
					Efficiency		Ligand
604	TSNQVAVLY	0.6559	2.7847	0.944	2.991	3.0758	yes
361	CVADYSVLY	0.5348	2.2705	0.9764	3.18	2.5759	yes
733	KTSVDCTMY	0.4908	2.084	0.9649	3.016	2.3795	yes
687	VASQSIIAY	0.3529	1.4986	0.9656	3.089	1.7978136	yes
136	CNDPFLGVY	0.2613	1.1095	0.69	2.45	1.3355	yes
261	GAAAYYVGY	0.2253	0.9568	0.7608	2.969	1.2194	yes
357	RISNCVADY	0.2106	0.8941	0.9292	3.394	1.2032	yes
285	ITDAVDCAL	0.235	0.9979	0.8708	0.79	1.168	yes
1237	MTSCCSCLK	0.226	0.9595	0.7525	0.479	1.0963	yes
50	STQDLFLPF	0.1974	0.8383	0.553	2.511	1.0468	yes
748	ECSNLLLQY	0.1413	0.6	0.5316	2.747	0.8171	yes

* Threshold was set at >0.75000. Bold shows the amino acids that were also predicted as antigenic sites.

**Table 6 biology-09-00296-t006:** CTL prediction from SARS-CoV-2 Main protease *.

Residue Number	Peptide Sequence	Predicted MHC Binding Affinity	Rescale Binding Affinity	C-Terminal Cleavage Affinity	TAP Transport Efficiency	Prediction Score	MHC Ligand
201	TVNVLAWLY	0.6255	2.6559	0.8852	2.957	2.9365	yes
110	QTFSVLACY	0.2625	1.1146	0.9725	2.998	1.4104	yes
153	DYDCVSFCY	0.2097	0.8905	0.9722	0.9722	1.1717	yes
93	TANPKTPKY	0.1676	0.7118	0.9755	2.676	0.9088	yes

***** Threshold was set at >0.75000. Bold shows the amino acids that were also predicted as antigenic sites.

**Table 7 biology-09-00296-t007:** Predicted CTL from SARS-CoV-2 Nsp12 RdRp *.

Residue Number	Peptide Sequence	Predicted MHC Binding Affinity	Rescale Binding Affinity	C-Terminal Cleavage Affinity	TAP Transport Efficiency	Prediction Score	MHC Ligand
738	DTDFVNEFY	0.7922	3.3634	0.8873	2.458	3.6194	yes
336	LSFKELLVY	0.3898	1.6552	0.9676	3.213	1.961	yes
27	STDVVYRAF	0.4019	1.7065	0.6174	2.4	1.9191	yes
859	FVSLAIDAY	0.3709	1.5746	0.7669	3.096	1.8444	yes
666	MVMCGGSLY	0.3637	1.5441	0.9482	3.008	1.8368	yes
758	LSDDAVVCF	0.3143	1.3345	0.9556	2.412	1.5985	yes
686	TTAYANSVF	0.2963	1.258	0.4772	2.663	1.4627	yes
762	AVVCFNSTY	0.2435	1.0339	0.9754	3.146	1.3375	yes
463	MCDIRQLLF	0.2518	1.0691	0.1005	2.436	1.206	yes
233	VVDSYYSLL	0.2332	0.9901	0.7134	0.834	1.1388	yes
700	VTANVNALL	0.2007	0.8523	0.9705	1.166	1.0562	yes
818	MLVKQGDDY	0.1793	0.7614	0.8328	3.079	1.0403	yes
823	GDDYVYLPY	0.1821	0.7733	0.8456	2.213	1.0108	yes
879	DVFHLYLQY	0.1677	0.7119	0.9529	3.013	1.0055	yes
876	EYADVFHLY	0.1624	0.6894	0.9603	2.953	0.9811	yes
230	GVPVVDSYY	0.1504	0.6386	0.9521	2.923	0.9276	yes
434	SVELKHFFF	0.1454	0.6176	0.9285	2.636	0.8886	yes
334	FVDGVPFVV	0.1739	0.7382	0.8437	0.191	0.8743	yes
645	CCSLSHRFY	0.1586	0.6732	0.274	2.91	0.8598	yes

* Threshold was set at >0.75000. Bold shows the amino acids that were also predicted as antigenic sites.

**Table 8 biology-09-00296-t008:** Predicted CTL from SARS-CoV-2 Nsp13 helicase *.

Residue Number	Peptide Sequence	Predicted MHC Binding Affinity	Rescale Binding Affinity	C-Terminal Cleavage Affinity	TAP Transport Efficiency	Prediction Score	MHC Ligand
57	VTDVTQLYL	0.4708	1.9988	0.6073	0.68	2.1239	yes
56	DVTDVTQLY	0.289	1.2271	0.9651	2.704	1.5071	yes
535	SSQGSEYDY	0.2761	1.1724	0.8149	2.847	1.437	yes
238	PTLVPQEHY	0.1794	0.7617	0.8719	2.595	1.0222	yes
448	IVDTVSALV	0.1991	0.8453	0.8977	0.133	0.9866	yes
574	CIMSDRDLY	0.1634	0.6937	0.1836	3.125	0.8775	yes
347	KVNSTLEQY	0.1391	0.5907	0.8156	2.971	0.8616	yes
245	HYVRITGLY	0.1102	0.4678	0.9598	3.009	0.7622	yes
85	ANGQVFGLY	0.1141	0.4845	0.9132	2.746	0.7588	yes
538	GSEYDYVIF	0.1401	0.5947	0.3528	2.203	0.7578	yes

* Threshold was set at >0.75000. Bold shows the amino acids that were also predicted as antigenic sites.

**Table 9 biology-09-00296-t009:** Conformational epitopes of SARS-CoV-2 Mpro.

No.	Residues	Number of Residues	Score
1	A:S1, A:G2, A:F3, A:A211, A:V212, A:I213, A:N214, A:G215, A:D216, A:R217, A:W218, A:F219, A:L220, A:N221, A:R222, A:F223, A:T224, A:T225, A:T226, A:L227, A:N228, A:D229, A:F230, A:N231, A:L232, A:V233, A:A234, A:M235, A:K236, A:Y237, A:Y239, A:E240, A:P241, A:L242, A:T243, A:Q244, A:D245, A:V247, A:D248, A:L250, A:G251, A:P252, A:S254, A:A255, A:Q256, A:T257, A:G258, A:I259, A:A260, A:V261, A:L262, A:D263, A:A266, A:S267, A:K269, A:E270, A:L271, A:L272, A:Q273, A:N274, A:G275, A:M276, A:N277, A:G278, A:R279, A:T280, A:I281, A:L282, A:G283, A:S284, A:A285, A:L286, A:C300, A:S301, A:G302	75	0.716

**Table 10 biology-09-00296-t010:** Conformational epitopes of SARS-CoV-2 S protein.

No.	Residues	Number of Residues	Score
1	A:L119, A:T120, A:K121, A:Y122, A:T123, A:D126, A:D135, A:E136, A:G137, A:N138, A:C139, A:D140, A:T141, A:K143, A:E144, A:I145, A:L146, A:V147, A:T148, A:Y149, A:N150, A:C151, A:C152, A:D153, A:D154, A:D155, A:Y156, A:F157, A:N158, A:K159, A:W162, A:Y163, A:N168, A:P169, A:D170, A:R173, A:V174, A:N177, A:L178, A:E180, A:R181, A:R183, A:Q184, A:A185, A:L187, A:K188, A:T189, A:V190, A:Q191, A:F192, A:C193, A:D194, A:A195, A:M196, A:R197, A:N198, A:A199, A:G200, A:I201, A:V202, A:G203, A:V204, A:L205, A:T206, A:D208, A:N209, A:Q210, A:D211, A:L212, A:N213, A:G214, A:N215, A:W216, A:Y217, A:D218, A:F219, A:G220, A:D221, A:F222, A:I223, A:Q224, A:T225, A:T226, A:P227, A:G228, A:S229, A:G230, A:V231, A:P232, A:V233, A:V234, A:A250, A:D284, A:K288, A:Y289	95	0.728
2	A:D269, A:L270, A:L271, A:K272, A:Y273, A:D274, A:F275, A:E277, A:E278, A:K281, A:T324, A:L329, A:V330, A:R331, A:K332, A:I333, A:F334, A:V335, A:D336, A:G337, A:V338, A:P339, A:F340, A:V341, A:V342, A:S343, A:T344, A:H355, A:N356, A:Q357, A:D358, A:V359, A:N360, A:L361, A:H362, A:S363, A:S364, A:R365, A:L366, A:S367, A:F368, A:K369, A:E370, A:L371, A:L372, A:V373, A:Y374, A:D377, A:P378, A:A379, A:M380, A:H381, A:A382, A:A383, A:S384, A:G385, A:N386, A:L387, A:L388, A:L389, A:D390, A:K391, A:R392, A:T393, A:A399, A:A400, A:L401, A:T402, A:N403, A:N404, A:V405, A:A406, A:F407, A:Q408, A:T409, A:V410, A:K411, A:P412, A:G413, A:N414, A:F415, A:N416, A:K417, A:D418, A:F419, A:Y420, A:D421, A:F422, A:A423, A:V424, A:S425, A:K426, A:G427, A:F428, A:F429, A:K430, A:E431, A:G432, A:S433, A:S434, A:V435, A:E436, A:L437, A:K438, A:H439, A:F440, A:F441, A:F442, A:A443, A:Q444, A:D445, A:G446, A:N447, A:C487, A:I488, A:N489, A:A490, A:N491, A:Q492, A:V493, A:D517, A:S518, A:M519, A:S520, A:Y521, A:E522, A:D523, A:Q524, A:D525, A:A526, A:L527, A:A529, A:Y530, A:T531, A:K532, A:R533, A:N534, A:V535, A:I536, A:Y546, A:A550, A:F594, A:Y595, A:G596, A:H599, A:N600, A:K603, A:S607, A:D608, A:V609, A:E610, A:N611, A:P612, A:H613, A:H642, A:T643, A:T644, A:C645, A:C646, A:S647, A:H650, A:G670, A:G671, A:T710, A:D711, A:G712, A:N713, A:K714, A:I715, A:A716, A:D717, A:K718, A:Y719, A:V720, A:R721, A:N722, A:L723, A:R726, A:C730, A:V737, A:D738, A:T739, A:D740, A:F741, A:N743, A:E744, A:K751, A:H752, A:N767, A:S768, A:T769, A:Y770, A:S772, A:Q773, A:G774, A:L775, A:V776, A:T801, A:E802, A:T803, A:D804, A:L805, A:T806, A:K807, A:G808, A:M818, A:L819, A:V820, A:K821, A:Q822, A:G823, A:D824, A:D825, A:Y826, A:V827, A:Y828, A:L829, A:P832, A:D833, A:P834, A:L838, A:G839, A:G841, A:C842, A:F843, A:V844, A:D845, A:D846, A:I847, A:V848, A:K849, A:T850, A:D851, A:G852, A:T853, A:L854, A:M855, A:I856, A:E857, A:F859, A:V860, A:A863, A:I864, A:A866, A:Y867, A:P868, A:L869, A:T870, A:K871, A:H872, A:P873, A:N874, A:Q875, A:E876, A:Y877, A:A878, A:D879, A:V880, A:F881, A:H882, A:L883, A:Y884, A:L885, A:Q886, A:Y887, A:I888, A:R889, A:K890, A:L891, A:H892, A:D893, A:E894, A:L895, A:T896, A:G897, A:H898, A:M899, A:L900, A:D901, A:M902, A:Y903, A:S904, A:V905, A:M906, A:L907, A:T908, A:N909, A:D910, A:N911, A:T912, A:S913, A:R914, A:Y915, A:W916, A:E917, A:P918, A:E919	297	0.719

**Table 11 biology-09-00296-t011:** Conformational epitopes from SARS-CoV-2 Nsp12 polymerase.

No.	Residues	Number of Residues	Score
1	A:D1139, A:P1140, A:L1141, A:Q1142, A:P1143, A:E1144, A:L1145, A:D1146	8	0.975
2	A:Y707, A:S708, A:N709, A:N710, A:S711, A:I712, A:A713, A:I714, A:P715, A:T716, A:N717, A:Q1071, A:K1073, A:N1074, A:F1075, A:T1076, A:T1077, A:A1078, A:P1079, A:A1080, A:I1081, A:C1082, A:H1083, A:D1084, A:G1085, A:K1086, A:A1087, A:H1088, A:F1089, A:P1090, A:R1091, A:E1092, A:G1093, A:V1094, A:F1095, A:V1096, A:S1097, A:N1098, A:G1099, A:T1100, A:H1101, A:W1102, A:F1103, A:V1104, A:T1105, A:Q1106, A:R1107, A:F1109, A:Y1110, A:E1111, A:P1112, A:Q1113, A:I1114, A:I1115, A:T1116, A:T1117, A:D1118, A:N1119, A:T1120, A:F1121, A:V1122, A:S1123, A:G1124, A:N1125, A:C1126, A:D1127, A:V1128, A:V1129, A:I1130, A:G1131, A:I1132, A:V1133, A:N1134, A:N1135, A:T1136, A:V1137, A:Y1138	77	0.845
3	A:L335, A:C336, A:P337, A:F338, A:G339, A:E340, A:V341, A:F342, A:N343, A:A344, A:T345, A:R346, A:F347, A:A348, A:S349, A:V350, A:Y351, A:A352, A:W353, A:N354, A:R355, A:K356, A:R357, A:I358, A:S359, A:N360, A:C361, A:V362, A:A363, A:D364, A:Y365, A:S366, A:V367, A:L368, A:Y369, A:N370, A:S371, A:A372, A:S373, A:F374, A:S375, A:T376, A:F377, A:K378, A:C379, A:Y380, A:L390, A:C391, A:F392, A:T393, A:N394, A:V395, A:Y396, A:A397, A:D398, A:S399, A:F400, A:V401, A:I402, A:R403, A:G404, A:D405, A:E406, A:V407, A:R408, A:Q409, A:I410, A:A411, A:P412, A:G413, A:Q414, A:T415, A:G416, A:K417, A:I418, A:A419, A:D420, A:Y421, A:N422, A:Y423, A:K424, A:L425, A:P426, A:D427, A:D428, A:F429, A:T430, A:G431, A:C432, A:V433, A:I434, A:A435, A:W436, A:N437, A:S438, A:N439, A:N440, A:L441, A:D442, A:S443, A:Y449, A:N450, A:Y451, A:L452, A:Y453, A:R454, A:P491, A:L492, A:Q493, A:S494, A:Y495, A:G496, A:F497, A:Q498, A:P499, A:T500, A:V503, A:G504, A:Y505, A:Q506, A:P507, A:Y508, A:R509, A:V510, A:V511, A:V512, A:L513, A:S514, A:F515, A:E516, A:L517, A:L518, A:H519, A:A520, A:P521, A:A522, A:T523, A:V524, A:C525, A:G526, A:P527, A:K528	142	0.799
4	A:F559, A:L560, A:P561, A:F562, A:Q563	5	0.789
5	A:F79, A:D80, A:N81, A:P82, A:V83, A:L84, A:P85, A:I100, A:I101, A:R102, A:G103, A:W104, A:I105, A:T108, A:T109, A:L110, A:D111, A:S112, A:K113, A:T114, A:Q115, A:S116, A:L117, A:L118, A:I119, A:V120, A:N121, A:N122, A:A123, A:T124, A:N125, A:V126, A:V127, A:I128, A:K129, A:V130, A:C131, A:E132, A:F133, A:Q134, A:F135, A:C136, A:N137, A:D138, A:P139, A:F140, A:L141, A:G142, A:E156, A:F157, A:R158, A:V159, A:Y160, A:S161, A:S162, A:A163, A:N164, A:N165, A:C166, A:T167, A:F168, A:E169, A:Y170, A:V171, A:S172, A:Q173, A:P174, A:F175, A:L176, A:T236, A:R237, A:F238, A:Q239, A:T240, A:L241, A:L242, A:A243, A:L244, A:H245, A:R246	80	0.756

**Table 12 biology-09-00296-t012:** Conformational epitopes from SARS-CoV-2 Nsp13 helicase.

No.	Residues	Number of Residues	Score
1	A:A1, A:V2, A:G3, A:A4, A:C5, A:L7, A:C8, A:N9, A:S10, A:Q11, A:T12, A:S13, A:L14, A:R15, A:C16, A:G17, A:F24, A:L25, A:C26, A:C27, A:K28, A:C29, A:C30, A:Y31, A:D32, A:V34, A:I35, A:S36, A:T37, A:S38, A:H39, A:K40, A:L41, A:V42, A:L43, A:S44, A:V45, A:N46, A:P47, A:Y48, A:V49, A:C50, A:N51, A:A52, A:P53, A:G54, A:C55, A:D56, A:V57, A:T58, A:D59, A:V60, A:T61, A:Q62, A:L63, A:Y64, A:L65, A:G66, A:G67, A:M68, A:S69, A:Y70, A:Y71, A:C72, A:K73, A:S74, A:H75, A:K76, A:P77, A:P78, A:I79, A:S80, A:F81, A:P82, A:L83, A:C84, A:A85, A:N86, A:G87, A:Q88, A:V89, A:F90, A:G91, A:L92, A:Y93, A:K94, A:N95, A:T96, A:C97, A:V98, A:G99, A:S100, A:D101, A:N102, A:V103, A:T104	96	0.761
2	A:D344, A:K345, A:F346	3	0.74
3	A:G150, A:I151, A:A152, A:T153, A:V154, A:R155, A:E156, A:V157, A:L158, A:S159, A:D160, A:R161, A:E162, A:L163, A:H164, A:L165, A:S166, A:W167, A:E168, A:V169, A:G170, A:K171, A:P172, A:R173, A:G184, A:Y185, A:R186, A:V187, A:T188, A:K189, A:N190, A:S191, A:K192, A:V193, A:Q194, A:I195, A:G203, A:D204, A:Y205, A:G206, A:D207, A:A208, A:V209, A:Y217, A:K218, A:L219, A:N220, A:V221, A:G222, A:D223, A:Y224, A:F225	52	0.738

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
