# Peer review of "A Putative Prophylactic Solution for COVID-19: Development of Novel Multiepitope Vaccine Candidate against SARS-COV-2 by Comprehensive Immunoinformatic and Molecular Modelling Approach"

_biology, 2020, doi:10.3390/biology9090296_

Round 1

Reviewer 1 Report

  1. Figure 3, A and B, should be improved
  2. Figure 3, C, D, E. and F labels are wrong. C should be radius of gyration (RoG), while D should be solvent-accessible surface area

Author Response

Reviewers' comments:

Reviewer 1:

  1. Figure 3, A and B, should be improved
  2. Figure 3, C, D, E. and F labels are wrong. C should be radius of gyration (RoG), while D should be solvent-accessible surface area

Thanks for the comment. We improved the color saturation tone of Figure 3A and 3B, and corrected the figure caption.

Reviewer 2 Report

The reviewer's question ahs been well answered. However there are still some things to modify clarify. 

 - If "there are numerous studies where both B cell and T cell epitopes are used for design of multiepitope vaccines against different pathogens" (as I know there are), just say that in the introduction. Is one of the main idead behind the rationbale of your work so should be explicitely mentioned it.

 - If Mpro and 3clPro are the same proten, clarify it. The reviewer didn't understand it reading the manuscripts so other people, not experts ins SARS-Cov-2, can have the same problem. Substitute "SARS-CoV-2 main 62 protease (Mpro) or chymotrypsin-like protease (3CLpro) ... " for something like "SARS-CoV-2 main 62 protease (Mpro), also known as chymotrypsin-like protease (3CLpro), ..."

- In figure S6 and S7 explain what is Chain A and Chain B. 

 - Check the tables in SI. When I opened the file tables S6 and S7 overlap between them.

- The authors have tabulated all the high score CTL/HTL epitopes. It seems that you have tabulated all the predicted indicating which are high scored, not only the high scored predicted epitopes. If this is the case just explain in the caption that thos with <-E are the top epitopes.

- PLease add this answer to the reviewer "ClusPro typically outputs 20-30 representative docking models. Each representative model is chosen from a cluster of docked models based on the scoring function it uses. We used the best conformation from the cluster holding the largest number of models. After MD, we performed MD clustering, which typically take the representative conformation from the largest cluster (within 1 Angstrom deviation). These cluster are generated based on the deviation over the course of total snapshots (in our case, after every 2ps which generated a total of 25000 snapshots). This analysis considers all meaningful conformations over the time of simulation." to the text. Is important to explain all the used procedure. Is also specially important to explain the limitations or possible pitfalls of the selected methods. That would help to people that want to follow your approach.

Author Response

The reviewer's question ahs been well answered. However there are still some things to modify clarify.

 - If "there are numerous studies where both B cell and T cell epitopes are used for design of multiepitope vaccines against different pathogens" (as I know there are), just say that in the introduction. Is one of the main idead behind the rationbale of your work so should be explicitely mentioned it.

 - If Mpro and 3clPro are the same proten, clarify it. The reviewer didn't understand it reading the manuscripts so other people, not experts ins SARS-Cov-2, can have the same problem. Substitute "SARS-CoV-2 main 62 protease (Mpro) or chymotrypsin-like protease (3CLpro) ... " for something like "SARS-CoV-2 main 62 protease (Mpro), also known as chymotrypsin-like protease (3CLpro), ..."

Thanks for the comment. We added “also known as” in its first appearance.

- In figure S6 and S7 explain what is Chain A and Chain B.

We added accordingly. Thanks

 - Check the tables in SI. When I opened the file tables S6 and S7 overlap between them.

We corrected accordingly.

- The authors have tabulated all the high score CTL/HTL epitopes. It seems that you have tabulated all the predicted indicating which are high scored, not only the high scored predicted epitopes. If this is the case just explain in the caption that thos with <-E are the top epitopes.

We carefully looked into it and arranged accordingly

- PLease add this answer to the reviewer "ClusPro typically outputs 20-30 representative docking models. Each representative model is chosen from a cluster of docked models based on the scoring function it uses. We used the best conformation from the cluster holding the largest number of models. After MD, we performed MD clustering, which typically take the representative conformation from the largest cluster (within 1 Angstrom deviation). These cluster are generated based on the deviation over the course of total snapshots (in our case, after every 2ps which generated a total of 25000 snapshots). This analysis considers all meaningful conformations over the time of simulation." to the text. Is important to explain all the used procedure. Is also specially important to explain the limitations or possible pitfalls of the selected methods. That would help to people that want to follow your approach.

Thanks for the suggestion. We added this information in the relevant position in methods section and reviewed it.

Reviewer 3 Report

The authors improved writing and added Materials and Methods section to improve organization. Additional results added in sections 3.10 Molecular docking of vaccine constructs with TLR3 and TLR4 and section 3.11. Molecular dynamics simulation for TLRs/MVC complex. Such additions provided more evidence to support this vaccine design. Several sentences were revised for improved readability. 

Detailed comments:

In section 3.7 Assessment of allergenicity and immunogenicity, additional evidence may be necessary. Current results indicate only one value: 0.4259. Since allergenicity is important for vaccine design, additional in vitro or in vivo experiments are necessary for a final vaccine product. Considering this paper is not a full vaccine product design, some relevant evidence can further improve this confidence on the results.

A few sentences can be edited. For example, 

The sentence on line 76: 

“Still an effective vaccine will require 12 to 18 months, if successful.”

can be edited to

“It will still requires 12 to 18 months to develop an effective vaccine.”

Author Response

This manuscript is a resubmission of an earlier submission. The following is a list of the peer review reports and author responses from that submission.

Round 1

Reviewer 1 Report

The authors have used in silico methods to design stable multiepitope vaccine construct (MVC) for COVID-19. I have following questions, suggestions and concerns about the paper:

  1. Lacks proper English. For example, in the abstract, authors wrote “In this urgency situation” . It should be changed to “The urgency of the situation requires…..”. There are many such sentences throughout the paper.
  2. It is important to cite most of the papers that have already shown to be potential drug candidates to treat COVID-19. For example, a.“An in silico approach for identification of novel inhibitors as potential therapeutics targeting COVID-19 main protease” https://www.tandfonline.com/doi/full/10.1080/07391102.2020.1776158“b. Computational Design of Potent Inhibitors for SARS-CoV-2's Main Protease”https://chemrxiv.org/articles/Computational_Design_of_Potent_Inhibitors_for_SARS-CoV-2_s_Main_Protease/12548003 c.“In Silicodesign and characterization of multiepitope vaccine for SARS-CoV2 from its Spike proteins” https://www.biorxiv.org/content/10.1101/2020.06.03.131755v1.full
  1. I don’t see any current advances in the field of COVID-19 vaccine in the introduction. Authors should write what the current advances in vaccine development.
  2. In the introduction: Authors wrote 4012 deaths with 113702 confirmed cases; in fact, it is way higher now. This needs to be fixed.
  3. In the Results, authors wrote “The 3D model was generated using various online servers, and a reliable model was used for docking and MD simulation studies”. It is important that authors write references for all these online servers.
  4. In Methods: Authors wrote “The most representative docked conformation from the largest cluster were used for further structural analyses.” What do they mean by representative docked conformation? How do they know the representative docked conformation? It should be mentioned.
  5. Most importantly, in Figure 2, Root-mean-square-deviation trajectory of MVC analyzed over a period of 50 ns MD simulations have been found to deviate quite a lot from initial structure, by about 9Ansgrom. This RMSD deviation is quite large and it most likely deviated from the real refined structure as there is no proof or validation about the structure. The structure obtained at 9 Angstrom RMSD may just represent a structure or conformation of the protein. Therefore, it is very important to validate their results with some sort of experimental data.
  6. Authors should also show RMSF, radius of gyration, solvent accessible surface area, principle component analysis etc. to show the stability of the complex and the simulation.
  7. Figure 3 should look nicer as the colors are very similar.

Reviewer 2 Report

Rehman and coworkers present a study aimed to find peptides, epitopes, succeptible to be part of a future Multiepitope vaccine. It is a potentially interesting study showing the capacity of computational tools to help in MVC construction. However it is just a preliminary test and further studies should be done to confirm this potential. The authors have to somehow explicitly comment it in the text, because they pointed out the possibility of that the interaction of a multiepitope vaccine construct designed through an integrated modelling approach may trigger innate and specific adaptive immunity by activating TLR signalling pathways and may produce a promising immune response against SARS-CoV-2, but is necessary to add the counterpart: that these results should be confirmed with more experiments.

A part of that the reviewer has some questions that would be interesting if the authors can answer:

1 - It would be nice if in the introduction the authors can elaborate a little bit more why B and T-cell epitope-based vaccines are under investigation, why they can be important, which are they objectives, etc.  And then explain why they try to use epitopes of SARS-CoV-2 Spike, Mpro, Nsp-12 polymerase, and Nsp13 helicase proteins and not 3CLPro that they seems to considered as important as the others, according to what they said in the paragraph above.

2 - For tables 5 to 8 the authors said that the selected molecules., that appear on the table, have a threshold >0.7500.  It would be nice to have a complete table with the values of all the molecules, in the SI, to be able to rationalize which Aa changes are important for a peptide sequence to become a CTL epitope.

3 - It would be nice to have a table with all the selected epitopes, indicating which are overlapped and which are high scored in CTL/HTL.

4 - It would be nice if the authors can elaborate a little bit more about why they choose the concrete adjuvants they select and also about why the GPGPG linkers and not other. In the discussion section they  said something more, respect to the results, but it still not enought.

5 - It would interesting if the authors can measure also the Rg and SASA over the MD. It will help to observe if the designed epitope fold too much or not and if there are regions able to bind the receptors.

6 - It is necessary that the authors perform replicas of the MD simulation to gain statistical insights that corroborates what is observed in a single simulation. 

7 - It would be interesting to select the top 3 o 5 models, ranked by ClusPro score, and perform MD simulation over all of them. Several times after MD when you allow a movement between the ligand (epitope)  and the target (receptor) you can observe the so-called inducing fit events that are consequence of the adaptation of ligand to the target and in the other way around. Usually after MD the docking ranking changes and the best docking pose is not the best anymore.

8 - It would be nice if the authors identify the binding mode, indicating the residues, involved in the epitope-receptors binding after docking and after MD of the docked complex. It will be an important information for future studies.

Reviewer 3 Report

This authors implemented a pipeline to design multiepitope a vaccine against SARS-COV2. They used a number of methods and platforms such as ElliPro and NetMHCII 2.2 Server to design the structure of the vaccine and test the allergenicity and stability of the vaccine. A simulation test was also implemented to test the immune response. As mentioned in the paper, in vitro and in vivo experiments are necessary for further research, since the overall research of this paper was completed by simulation.

The overall pipeline of vaccine design was relative straightforward. The design was based on epitopes of B-cell and T-cell to enhanced the immunity of the human body, thus allergenic test was important. The paper used two platforms for the allergenic test; secondary metabolism may still be an issue for unpredicted reactions.
